

# Bibliometric analysis of research on the role of intestinal microbiota in obesity

Haiqiang Yao[1,2,3,*], Jin-Yi Wan[1,2,4,*], Chong-Zhi Wang[2,3], Lingru Li[1], Ji Wang[1], Yingshuai Li[1], Wei-Hua Huang[2,3], Jinxiang Zeng[2,3], Qi Wang[1] and Chun-Su Yuan[2,3]

[1] Beijing University of Chinese Medicine, Beijing, China
[2] Tang Center for Herbal Medicine Research, The University of Chicago, Chicago, IL, United States of America
[3] Department of Anesthesia and Critical Care, The University of Chicago, Chicago, IL, United States of America
[4] School of Pharmacy, Jiangsu University, Zhenjiang, China
[*] These authors contributed equally to this work.

## ABSTRACT

**Background**. Obesity is a key public health problem. The advancement of gut microbiota research sheds new light on this field. This article aims to present the research trends in global intestinal microbiota studies within the domain of obesity research.

**Methods**. Bibliographic information of the publications on intestinal microbiota and obesity was retrieved from the Scopus database, and then analyzed by using bibliometric approaches.

**Results**. A total of 3,446 references were retrieved; the data indicated a steady growth and an exponential increase in publication numbers. The references were written in 23 different languages (93.8% in English). A number of 3,056 English journal papers were included in the further analyses. Among the 940 journals, the most prolific ones were *PLOS ONE*, *Scientific Reports*, and *British Journal of Nutrition*. North America and Europe were the highest publication output areas. The US (995 publications) ranked first in the number of publications, followed by the China (243 publications) and France (242 publications). The publication numbers were significantly correlated with gross domestic product (GDP), human development index (HDI), and population number (PN). International collaboration analysis also shows that most of the collaborations are among developed countries.

**Discussion**. This comprehensive bibliometric study indicates that gut microbiota is a significant topic in the obesity research. The structured information may be helpful in understanding research trends, and locating research hot spots and gaps in this domain.

Corresponding author
Qi Wang, wangqi710@126.com

## INTRODUCTION

Obesity, marked by the excess body fat accumulation, is a major public health crisis. This medical condition has almost become a global epidemic and is progressing rapidly (*Barbieri et al., 2017*; *D'Souza et al., 2017*). Among factors, intestinal microbiota can influence

the whole-body metabolism by affecting the host's energy homeostasis (*De Vadder & Mithieux, 2018*; *Omari-Siaw et al., 2016*). There are ample studies showing that probiotics and prebiotics, or other dietary substances, can be used to alleviate obesity through the modulation of the host's intestinal microbiota (*Bird et al., 2017*; *Li et al., 2017a*; *Li et al., 2017b*). However, to date, the role of intestinal microbiota in the formation and progression of obesity has not been systematically presented using bibliometric analysis.

Bibliometric analysis can be applied to track the developing trends, access the influences of publications, and compare the academic performance between different regions of a certain research field. This bibliometric technique has been applied in the domain of obesity; one publication investigated studies from 1998–2007 using data from PubMed (*Vioque et al., 2010*), and another study analyzed the longitudinal trends from 1993 to 2012 with Scopus as the data source (*Khan et al., 2016*). However, to our knowledge thus far, no bibliometric studies have assessed intestinal microbiota and obesity research at the global level.

The aim of this study was to present an overview of the research trends on intestinal microbiota and obesity up to now and shed new light on future research directions. We investigated the growth and citation of publications, active authors, countries and institutions, international collaboration, and the frequency of terms through a bibliometric analysis.

## METHODS

### Search strategy

Bibliometric data can be acquired through various search engines. In this study, the Scopus database was selected to perform the literature search for all published articles on enteric microbiota and obesity. It was justifiable to use Scopus as our data source to retrieve abstracts, citations and other bibliometric data given that it has wider resources and is consistently more accurate than other alternatives such as PubMed, Web of Science and Google Scholar (*Choudhri et al., 2015*; *Falagas et al., 2008*; *Kulkarni et al., 2009*).

The Scopus database was searched from its inception to December 31, 2017 with no language limitation. The synonyms for gut microbiota and obesity were included in the search strategy. The keywords pertaining to gut microbiota were: gastrointestinal microbiomes, gut microflora, gut microbiota, gastrointestinal flora, gut flora, gastrointestinal microbiota, gut microbiome, gastrointestinal microflora, intestinal microbiome, intestinal microbiota, intestinal microflora, and intestinal flora. The keywords regarding obesity were: obesity, corpulence, fatness, and overweight. The two sets of keywords were searched with the AND logic in the Article title/ Abstract/ Keywords fields. The final search query was built like this:

((TITLE-ABS-KEY ( gastrointestinal AND microbiomes ) OR TITLE-ABS-KEY ( gut AND microflora ) OR TITLE-ABS-KEY ( gut AND microbiota ) OR TITLE-ABS-KEY ( gastrointestinal AND flora ) OR TITLE-ABS-KEY ( gut AND flora ) OR TITLE-ABS-KEY ( gastrointestinal AND microbiota ) OR TITLE-ABS-KEY ( gut AND microbiome ) OR TITLE-ABS-KEY ( gastrointestinal AND microflora ) OR TITLE-ABS-KEY ( intestinal AND microbiome ) OR TITLE-ABS-KEY ( intestinal AND microbiota ) OR TITLE-ABS-KEY ( intestinal AND microflora ) OR TITLE-ABS-KEY ( intestinal AND flora ) ) AND

PUBYEAR <2018 ) AND ( ( TITLE-ABS-KEY ( obesity ) OR TITLE-ABS-KEY ( corpulence ) OR TITLE-ABS-KEY ( fatness ) OR TITLE-ABS-KEY ( overweight ) ) AND PUBYEAR <2018 ).

All of the citation information, bibliographical information, abstract and keywords, funding details and other information of the retrieved publications was exported with CSV format through the University of Chicago Library's access, for the further data processing.

## Data analysis

All bibliometric information was exported into CSV format from the Scopus database. Microsoft Excel was applied to for sorting and to perform statistical procedures. Top prolific authors, countries, journals, institutions and most cited papers were ranked according to the standard competition ranking (SCR). Data visualization was conducted using the VOSviewer technique to create scientific landscapes and networks based on the citation frequency, countries, journals, authors and other information (*Van Eck & Waltman, 2010*). GunnMap 2 (http://lert.co.nz/map/) was used to generate the world map to show the publication distribution.

Some bibliometric indicators were applied in the analysis. The strength of publications included was assessed by impact factor (IF) obtained from the latest Journal Citation Report (2016) published by Thomson Reuters. Bradford's Law was used as a scattering index to reveal the distribution of the scientific literature in a particular discipline. Bradford proposed a model of concentric zones of productivity, termed as Bradford zones, with decreasing densities of literature. Each of these zones would contain a similar number of articles, but the number of journals in which these articles are published would increase on passing from one zone to another (*Bradford, 1948*). This model permits the identification of journals that are most widely used or have the greatest weightage in a given field of scientific production.

GraphPad Prism version 6.0c (San Diego, CA, USA) was used to conduct statistical analysis. Pearson's correlation analysis was used to investigate relationships between the publication numbers of different countries/regions and some related variables, such as gross domestic product (GDP), GDP per capita (purchasing power parity, PPP), etc. A $P$ value of <0.05 was considered statistically significant.

# RESULTS

## Publications analysis based on numbers and citations

The applied strategy yielded a total of 3,446 publications. The first article on gut microbiota and obesity was published in 1968, and the annual publication numbers were stable in the following nearly 40 years till 2004. A dramatic growth was observed in the last 15 years from 2003 to 2017, and the number of annual documents during this period showed an exponential growth trend ($y = 0e^{0.4263x}$, $R^2 = 0.94$). The specific numbers of annual documents and accumulated documents are shown in Fig. 1A. The highest number of annual publications was seen in 2017, totaling 702 publications. The publication numbers of obesity research (Fig. 1B) can serve as a background frame to present a better understanding of the growth trend of intestinal microbiota in obesity research. As shown in

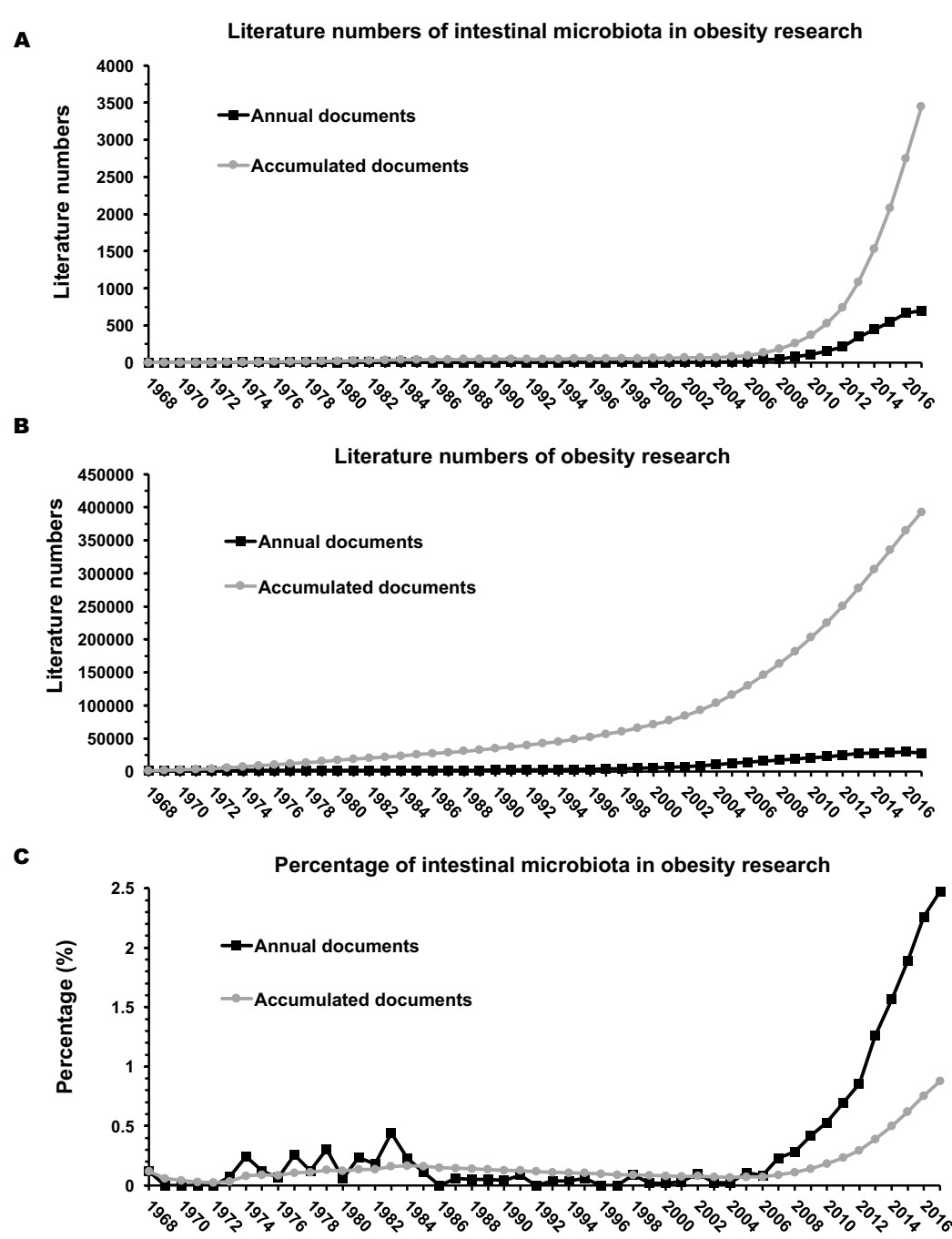

**Figure 1** Annual and accumulated publications of intestinal microbiota and obesity (A), obesity research (B), and the percentage of intestinal microbiota related publications in the obesity research (C).

Fig. 1C, the percentage of intestinal microbiota related publications in the obesity research was increasing gradually, especially after the year of 2006.

In terms of document type, the majority of the retrieved papers were research articles ($n = 1826$, 53.0%), followed by review articles ($n = 1110$, 32.2%) and book chapters

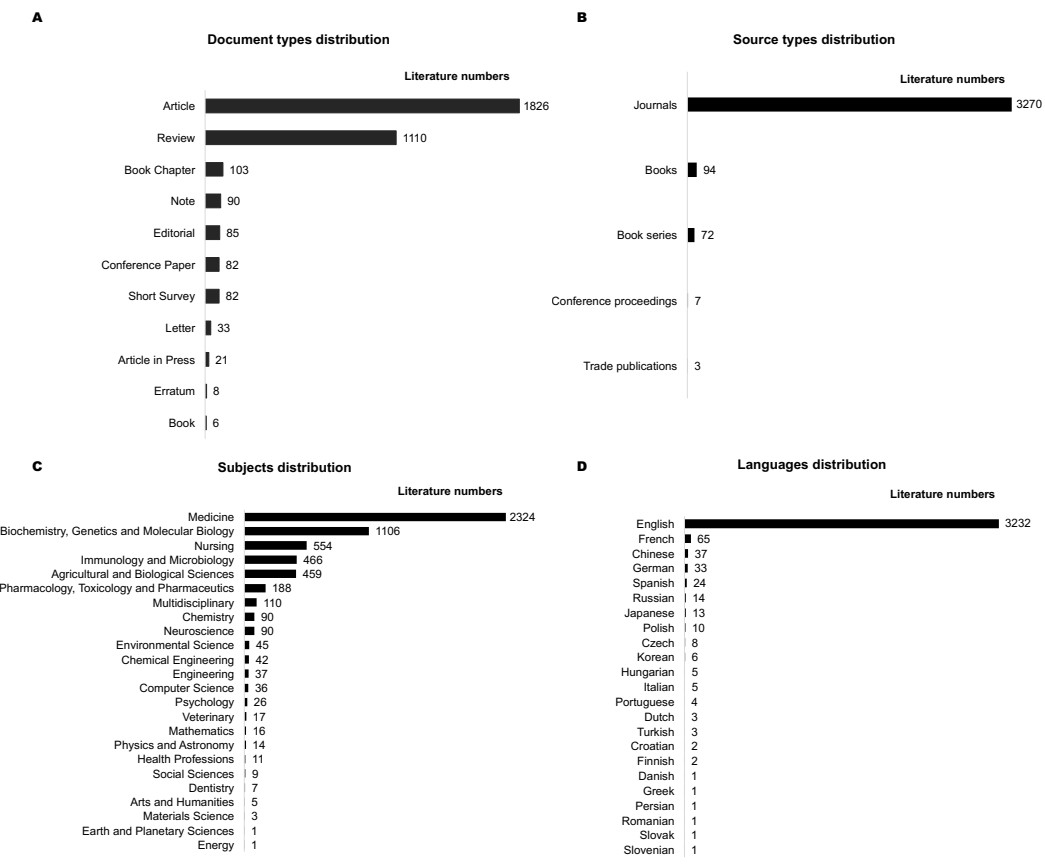

**Figure 2  General information regarding the 3,446 publications retrieved on gut microbiota and obesity.** (A) Document types distribution; (B) Source types distribution; (C) Subjects distribution; (D) Languages distribution.

($n = 103$, 3.0%) (Fig. 2A). The primary source of the publications was journals (Fig. 2B). As shown in Fig. 2C, most of the retrieved documents belong to medical subjects ($n = 2,324$, 67.4%), followed by Biochemistry, Genetics and Molecular Biology ($n = 1,106$, 32.1%), Nursing ($n = 554$, 16.1%), Immunology and Microbiology ($n = 466$, 13.5%), and Agricultural and Biological Sciences ($n = 459$, 13.3%). The retrieved papers were written in 23 different languages, mainly in English ($n = 3232$, 93.8%), followed by French ($n = 65$, 1.9%), Chinese ($n = 37$, 1.1%), German ($n = 33$, 1.0%), and Spanish ($n = 24$, 0.7%).

All of the retrieved 3,446 papers were cited a total of 141,918 times. A total of 2,858 (82.9%) articles had at least one citation while 588 (17.1%) articles had no citations.

Among the total 3,446 documents there were multiple source types (Fig. 2B) that were written in different languages (Fig. 2D). Considering the heterogeneity of the documents may add background noises to the study, we only included the journal articles written in English to perform the further analyses. According to this inclusion criteria, a total of 390 documents (books, book series, conference proceedings and other types) were excluded. 3,056 English papers were included for the subsequent analyses.

## Publications analysis based on top cited articles

Top 20 cited papers are listed in Table 1 (*Bäckhed et al., 2004*; *Bäckhed et al., 2005*; *Bäckhed et al., 2007*; *Cani et al., 2007*; *Cani et al., 2008*; *Cani et al., 2009*; *Henao-Mejia et al., 2012*; *Kau et al., 2011*; *Le Chatelier et al., 2013*; *Ley et al., 2005*; *Ley et al., 2006*; *O'Hara & Shanahan, 2006*; *Qin et al., 2010*; *Ridaura et al., 2013*; *Tremaroli & Bäckhed, 2012*; *Turnbaugh et al., 2008*; *Turnbaugh et al., 2009a*; *Turnbaugh et al., 2006*; *Turnbaugh et al., 2009b*; *Vijay-Kumar et al., 2010*). The highest citation number was 3,961 for the article entitled "An obesity-associated gut microbiome with increased capacity for energy harvest" (*Turnbaugh et al., 2006*). Among the top 20 highly cited papers, eight were published in *Nature*, followed by *Science* and *Proceedings of the National Academy of Sciences of the United States of America* (three papers), *Diabetes* (two papers). As the first author, Turnbaugh PJ published four papers among the top 20 cited papers, followed by Bäckhed F. and Cani P.D. (three papers), then by Ley RE (two papers). Among the top 20 cited papers, Gordon, J.I. participated 12 papers, followed by Ley R.E. (six papers) and Bäckhed F. (five papers); they were the most productive of the highest quality authors.

## Publications analysis based on authors

A total of 10,648 authors contributed to the publication of all the 3,056 papers included. The number of authors for a single document, also known as the transience index, is 8,224, accounting for 77.2% of all the authors. There are 85 authors who published more than ten articles in this field. Cani, P.D. was the most productive author in this field with 76 publications, the following were Delzenne, N.M. (52 publications) and Bäckhed, F. (48 publications). Details of the top 20 most prolific authors are presented in Fig. S1A, and the co-authorship analysis is shown in Fig. S2.

A total of 9,590 authors have been cited at least once, accounting for 91.7% of the total 10,462 authors, 6,064 authors had citation numbers at least 10 (58.0%), and 1,597 authors have been cited at least 100 times (15.3%). Gordon JI was the most influential author with the greatest citation number of 25,073, followed by Ley R.E. (17,658), and Bäckhed F. (15,518); details of citation analysis are represented in Fig. 3. Among the top 20 most cited authors, Batoo, J-M, Levenez, F and Renault, P had the greatest citation numbers per publication (1,303.8), followed by Tap, J (1,297.5) and Parkhill, J (1,291.5), more details are shown in Fig. S1B.

## Publications analysis based on countries/regions

The geographical distribution of publications involved 79 countries/regions over six continents (Fig. 4). There were 20 countries that just published only one article, and 46 countries that published at least five articles. The top 20 most productive countries are shown in Fig. S1C; the US ranked first with 995 publications, followed by China (243 publications), France (242 publications), United Kingdom (223 publications), and Italy (196 publications). However, the GDP-adjusted ranking according to their productivity scores was different, the most prolific countries were Finland, Denmark, Ireland, Belgium, and Swaziland (Fig. S1D). Publication productivity scores of countries/regions were calculated by dividing the number of publications by their GDP and then multiplying the

**Table 1  Top 20 cited articles on gut microbiota and obesity from inception to 2017.**

| SCR | Article | Title | Year | Source title | Cited by | IF |
|---|---|---|---|---|---|---|
| 1st | Turnbaugh et al. (2006) | An obesity-associated gut microbiome with increased capacity for energy harvest | 2006 | Nature | 3,961 | 40.137 |
| 2nd | Qin et al. (2010) | A human gut microbial gene catalogue established by metagenomic sequencing | 2010 | Nature | 3,724 | 40.137 |
| 3rd | Turnbaugh et al. (2009a) | A core gut microbiome in obese and lean twins | 2009 | Nature | 3,072 | 40.137 |
| 4th | Ley et al. (2006) | Microbial ecology: Human gut microbes associated with obesity | 2006 | Nature | 3,023 | 40.137 |
| 5th | Bäckhed et al. (2004) | The gut microbiota as an environmental factor that regulates fat storage | 2004 | PNAS | 2,350 | 9.661 |
| 6th | Ley et al. (2005) | Obesity alters gut microbial ecology | 2005 | PNAS | 2,292 | 9.661 |
| 7th | Bäckhed et al. (2005) | Host-bacterial mutualism in the human intestine | 2005 | Science | 2,174 | 37.205 |
| 8th | Cani et al. (2007) | Metabolic endotoxemia initiates obesity and insulin resistance | 2007 | Diabetes | 1,986 | 8.684 |
| 9th | Cani et al. (2008) | Changes in gut microbiota control metabolic endotoxemia-induced inflammation in high-fat diet-induced obesity and diabetes in mice | 2008 | Diabetes | 1,521 | 8.684 |
| 10th | Tremaroli & Bäckhed (2012) | Functional interactions between the gut microbiota and host metabolism | 2012 | Nature | 1,198 | 9.661 |
| 11st | Turnbaugh et al. (2009b) | The effect of diet on the human gut microbiome: a metagenomic analysis in humanized gnotobiotic mice | 2009 | Science Translational Medicine | 1,091 | 16.761 |
| 12nd | Bäckhed et al. (2007) | Mechanisms underlying the resistance to diet-induced obesity in germ-free mice | 2007 | PNAS | 1,081 | 9.661 |
| 13rd | Turnbaugh et al. (2008) | Diet-induced obesity is linked to marked but reversible alterations in the mouse distal gut microbiome | 2008 | Cell Host and Microbe | 1,046 | 14.946 |
| 14th | Vijay-Kumar et al. (2010) | Metabolic syndrome and altered gut microbiota in mice lacking toll-like receptor 5 | 2010 | Science | 996 | 37.205 |
| 15th | O'Hara & Shanahan (2006) | The gut flora as a forgotten organ | 2006 | EMBO Reports | 949 | 8.568 |
| 16th | Cani et al. (2009) | Changes in gut microbiota control inflammation in obese mice through a mechanism involving GLP-2-driven improvement of gut permeability | 2009 | Gut | 903 | 16.658 |
| 17th | Le Chatelier et al. (2013) | Richness of human gut microbiome correlates with metabolic markers | 2013 | Nature | 900 | 40.137 |
| 18th | Kau et al. (2011) | Human nutrition, the gut microbiome and the immune system | 2011 | Nature | 885 | 40.137 |
| 19th | Henao-Mejia et al. (2012) | Inflammasome-mediated dysbiosis regulates progression of NAFLD and obesity | 2012 | Nature | 884 | 40.137 |
| 20th | Ridaura et al. (2013) | Gut microbiota from twins discordant for obesity modulate metabolism in mice | 2013 | Science | 882 | 37.205 |

**Notes.**

SCR, standard competition ranking. Equal items were given the same ranking number, and then a gap is left in the ranking numbers; PNAS, Proceedings of the National Academy of Sciences of the United States of America; IF, impact factor.

[a]Data extracted from Journal Citation Reports, Thomson Reuters, 2016.

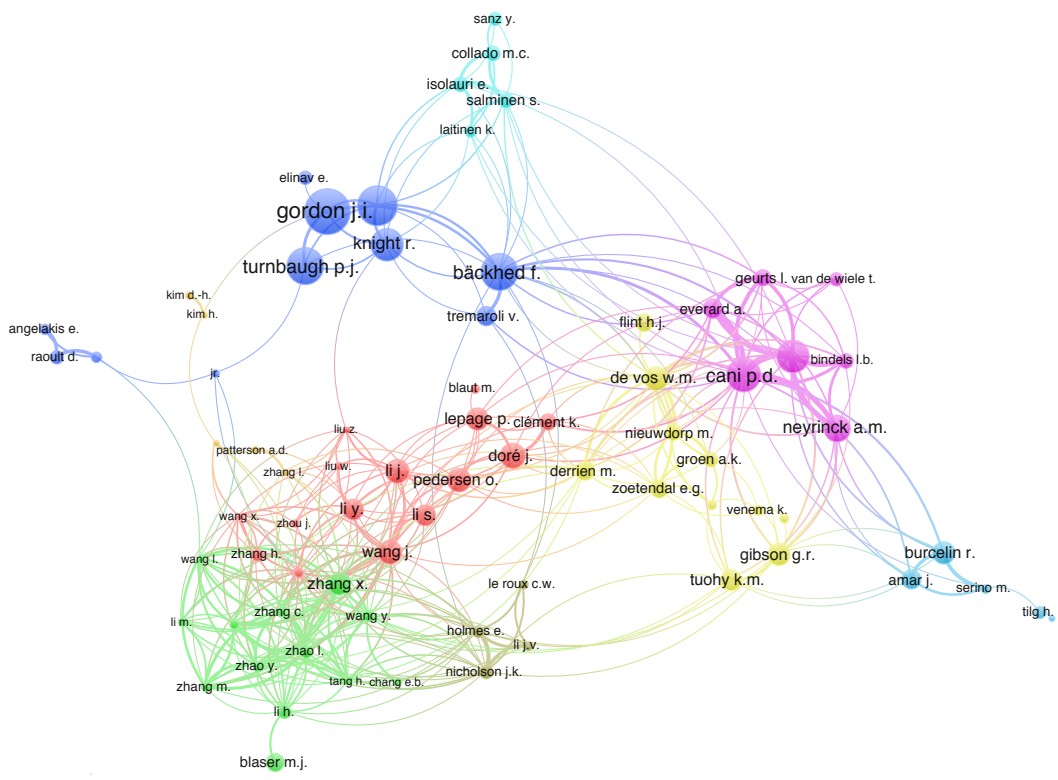

**Figure 3  Citation analysis of 85 authors with at least 10 publications.** Thicker lines indicate stronger collaborations. Authors represented with larger circle size or font size had relatively more citations.

result with by 1E + 12. As shown in the Fig. S1E, the most productive continents according to the publication numbers were Europe, North America, and Asia.

In order to achieve a better understanding of the diverse publication numbers in different countries/regions, the relationship between the publication numbers and multiple factors for each country was investigated. The analyzed factors were population number (PN), gross domestic product (GDP), gross domestic product per capita (purchasing power parity, PPP), and human development index (HDI). Significant correlation was found between the publication number of intestinal microbiota in obesity research and GDP, HDI, and PN (Fig. 5). GDP demonstrated the highest correlation with the publication number ($r = 0.89$, $p < 0.0001$), followed by HDI ($r = 0.32$, $p < 0.001$) and PN ($r = 0.27$, $p < 0.05$). GDP was the most important factor that could benefit the publication productivity. Therefore, this analysis can interpret the reason why the US and China were the most prolific countries in this domain.

Analysis of citation counts for countries showed that the US had the most citations followed by France, UK, Belgium and the Netherlands (Fig. S3). The international collaboration analysis based on these countries is shown in a network visualization map (Fig. S4). As is indicated in the collaboration analysis, the United States (US) had collaborations with other 38 countries and followed by Italy (35 collaboration links), Spain (33 collaboration links), United Kingdom (UK) (32 collaboration links), and Germany

![PeerJ]

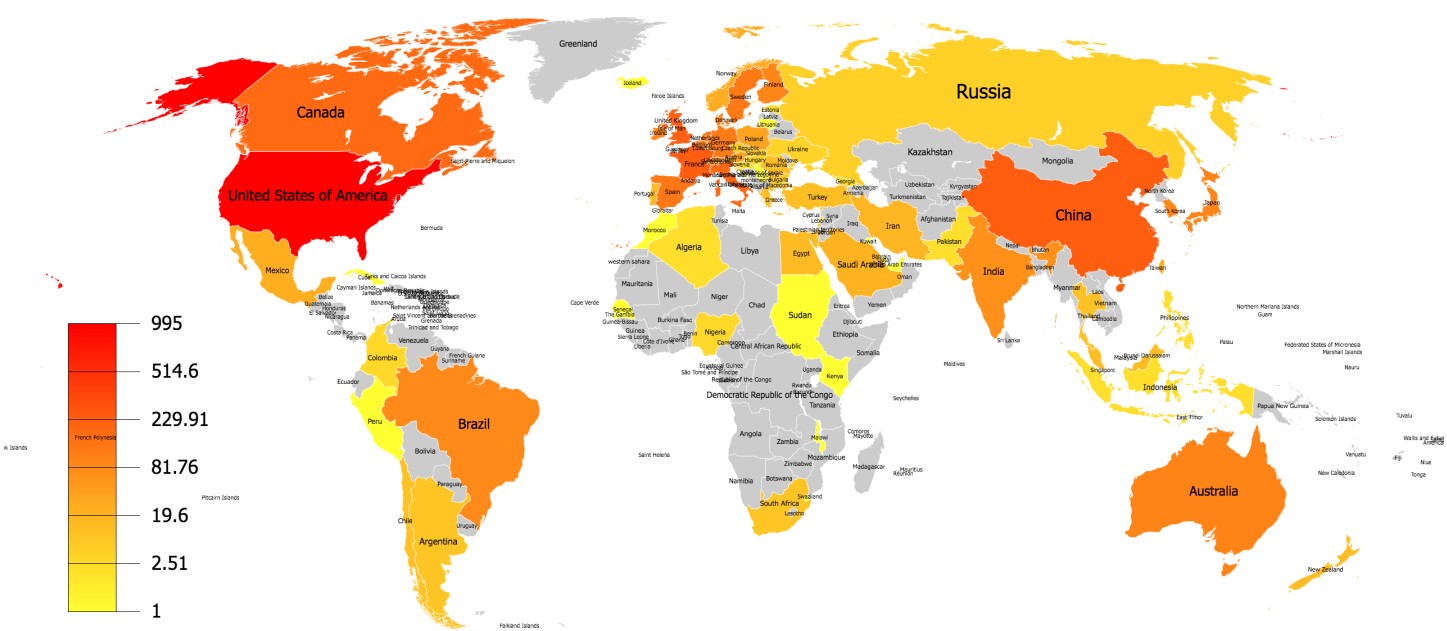

**Figure 4** Geographical distribution map of publications on intestinal microbiota in obesity research.

(31 collaboration links). For the US, collaboration was mostly with China (relative link strength = 64) and Canada (relative link strength = 45).

## Publications analysis based on institutions

Prolific institutions in publishing papers on gut microbiota and obesity are presented in Table 2. The most active institution was *University College Cork* in Ireland (122 publications), followed by *The Institut national de la santé et de la recherche médicale* (Inserm) in France (109 publications), and *Universite Catholique de Louvain* in Belgium (95 publications). A total of 160 institutions published at least 10 articles. Among the top 20 most active institutions, 14 are in Europe, four and in North America, and two are in China. A total of 79 organizations have been cited at least 1,000 times, the citation analysis was shown in Fig. S5. *The Department of Chemistry and Biochemistry, University of Colorado* obtained the highest citation number (7,930 citations), and followed by *Center for Genome Sciences, Washington University in St. Louis* (5,032 citations) and *Danone Research*, in France (4,652 citations).

## Publications analysis based on journals

All the retrieved documents were published in 940 different journals. The top 20 active journals in publishing articles on gut microbiota and obesity are shown in Table 3. The most prolific journal in this field was *PLOS ONE* (106 publications), followed by *Scientific Reports* (46 publications), and *British Journal of Nutrition* (43 publications). The total number of articles published in top 20 journals was 728, accounting for 23.8% of total retrieved documents. Citations analysis of the included 143 journals with at least five publications indicated that *Nature* has the highest citation numbers

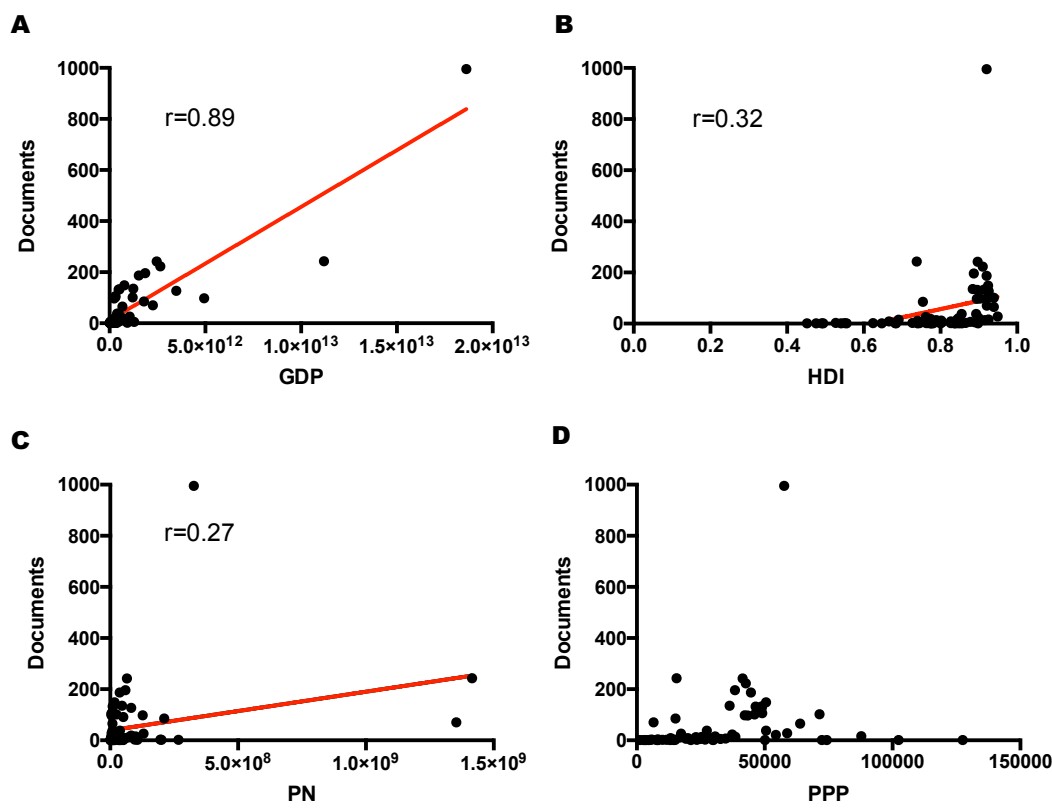

**Figure 5  Correlations between the publication number of bariatric intestinal microbiota in obesity research and (A) GDP ($r = 0.89$, $p < 0.0001$), (B) HDI ($r = 0.32$, $p < 0.001$), (C) PN ($r = 0.27$, $p < 0.05$), and (D) PPP ($p = 0.08$).** Abbreviations: GDP, gross domestic product; HDI, human development index; PN, population number; PPP, purchasing power parity (gross domestic product per capita).

($N = 22,139$), followed by *Proceedings of The National Academy of Sciences of The United States of America* ($N = 10,195$), and *PloS One* ($N = 5,117$) (Fig. 6). Regarding the academic journals publishing articles on gut microbiota and obesity, the Bradford model was applied. The division of Bradford's zones in this study is shown in Table 4. A total of 940 journals were involved; the average number of articles in each zone was 305.6. The first zone included six journals: *Plos One, Scientific Reports, British Journal of Nutrition, Nutrients, Gut Microbes,* and *Cell Metabolism.*

## Publications analysis based on terms frequency

A density visualization map of most frequently encountered terms is shown in Fig. 7. This analysis was performed based on the terms extracted from the title and abstract fields of retrieved publications; a number of 195 terms met the threshold with a minimum number of occurrences as 100. The term with the highest frequency was obesity ($N = 4,677$), followed by gut microbiota ($N = 3,437$), and diet ($N = 2,267$).

**Table 2  Top 20 prolific institutions in publishing papers on gut microbiota and obesity.**

| SCR | Institution | Country | Documents | % N = 3,056 |
|---|---|---|---|---|
| 1st | University College Cork | Ireland | 122 | 4.0 |
| 2nd | Inserm | France | 109 | 3.6 |
| 3rd | Universite Catholique de Louvain | Belgium | 95 | 3.1 |
| 4th | Kobenhavns Universitet | Denmark | 87 | 2.8 |
| 5th | Goteborgs Universitet | Sweden | 74 | 2.4 |
| 6th | Wageningen University and Research Centre | The Netherlands | 60 | 2.0 |
| 7th | INRA Institut National de La Recherche Agronomique | France | 53 | 1.7 |
| 7th | The Wallenberg Laboratory | Sweden | 53 | 1.7 |
| 9th | Imperial College London | UK | 49 | 1.6 |
| 10th | VA Medical Center | US | 46 | 1.5 |
| 11st | Harvard Medical School | US | 45 | 1.5 |
| 11st | Helsingin Yliopisto | Finland | 45 | 1.5 |
| 13rd | CNRS Centre National de la Recherche Scientifique | France | 41 | 1.3 |
| 14th | Novo Nordisk Foundation | Denmark | 40 | 1.3 |
| 14th | Consejo Superior de Investigaciones Científicas | Spain | 40 | 1.3 |
| 16th | Chinese Academy of Sciences | China | 39 | 1.3 |
| 16th | Shanghai Jiao Tong University | China | 39 | 1.3 |
| 18th | University of Calgary | Canada | 37 | 1.2 |
| 19th | Maastricht University | The Netherlands | 36 | 1.2 |
| 20th | University of Chicago | US | 33 | 1.1 |

## DISCUSSION

Obesity is a leading cause of preventable death in the US. Obesity induces a number of health problems, both independently and in association with other diseases, such as coronary heart disease, type 2 diabetes, and an increased incidence of several forms of cancer (*Feng et al., 2015*; *Ozturk, 2017*; *Wang et al., 2015*). Some variables such as host genetics, sedentary lifestyle and high-fat diet are identified as etiological factors of obesity (*Cheng et al., 2016*; *Kobyliak et al., 2016*), however, the in-depth pathogenesis that accounts for the development of obesity has yet to be disclosed.

Experimentally, model organisms provide an important approach for understanding the cause behind different disorders. Animal models approximate some human diseases, such as diabetes and obesity, and could reproduce many these medical conditions. As is indicated in recent enteric microbiota studies, the community of microorganisms residing in the gastrointestinal tract, is playing a major role in the onset and development of obesity (*Duranti et al., 2017*; *Kvit & Kharchenko, 2017*). The roles of microbiota can be explored within the constraints of particular animal model systems, although standard models of inbred mice are limited by their uncontrolled microbiome diversity (*Cho & Blaser, 2012*).

Intestinal microbiota is involved in the pathogenesis of obesity through various of pathways. Gut microbiota can influence the whole-body metabolism by affecting the host's energy homeostasis through the mechanism of adjusting the quantity of effector molecules to finally regulate the fat storage in adipocytes (*Bai, Zhu & Dong, 2016*;

**Table 3  Top 20 prolific journals in publishing papers on gut microbiota and obesity.**

| SCR | Journals | Documents | % N = 3,056 | IF |
|------|----------|-----------|-------------|------|
| 1st | Plos One | 106 | 3.5 | 2.806 |
| 2nd | Scientific Reports | 46 | 1.5 | 4.259 |
| 3rd | British Journal of Nutrition | 43 | 1.4 | 3.706 |
| 4th | Nutrients | 42 | 1.4 | 3.55 |
| 5th | Gut Microbes | 40 | 1.3 | NA |
| 5th | Cell Metabolism | 37 | 1.2 | 18.164 |
| 7th | Frontiers in Microbiology | 36 | 1.2 | 4.076 |
| 7th | World Journal of Gastroenterology | 36 | 1.2 | 3.365 |
| 9th | Gastroenterology | 33 | 1.1 | 18.392 |
| 10th | Nature | 32 | 1.0 | 40.137 |
| 11st | Beneficial Microbes | 30 | 1.0 | 2.923 |
| 11st | Molecular Nutrition and Food Research | 30 | 1.0 | 4.323 |
| 11st | Nature Reviews Gastroenterology and Hepatology | 30 | 1.0 | 13.678 |
| 14th | Gut | 29 | 0.9 | 16.658 |
| 14th | International Journal of Obesity | 29 | 0.9 | 5.487 |
| 16th | American Journal of Clinical Nutrition | 27 | 0.9 | 6.926 |
| 16th | Current Opinion in Clinical Nutrition and Metabolic Care | 27 | 0.9 | 4.023 |
| 18th | Nature Reviews Endocrinology | 26 | 0.9 | 18.318 |
| 19th | Journal of Nutrition | 25 | 0.8 | 4.145 |
| 20th | Journal of Nutritional Biochemistry | 24 | 0.8 | 4.518 |

**Notes.**

Abbreviation: NA, not available.

**Table 4  Distribution of the journals in Bradford's zones.**

| Bradford's Zones | Number of Journals | % Journals | Number of articles | Bradford's multiplier |
|------------------|--------------------|-----------|--------------------|-----------------------|
| 1 | 6 | 0.6 | 314 | |
| 2 | 10 | 1.1 | 312 | 1.67 |
| 3 | 15 | 1.6 | 304 | 1.50 |
| 4 | 24 | 2.6 | 300 | 1.60 |
| 5 | 35 | 3.7 | 302 | 1.46 |
| 6 | 53 | 5.6 | 305 | 1.51 |
| 7 | 88 | 9.4 | 305 | 1.66 |
| 8 | 139 | 14.8 | 304 | 1.58 |
| 9 | 265 | 28.2 | 305 | 1.91 |
| 10 | 305 | 32.4 | 305 | 1.15 |

Total number of journals = 940

Average number of articles in each zone = 305.6

*De Clercq et al., 2016*). Besides, gut saccharolytic microorganisms can degrade complex dietary glycans, which humans cannot utilize directly, and then provide the host with a variety of metabolites. This process may exert extensive influence on the glucose, cholesterol, and lipid metabolism (*Duranti et al., 2017*).

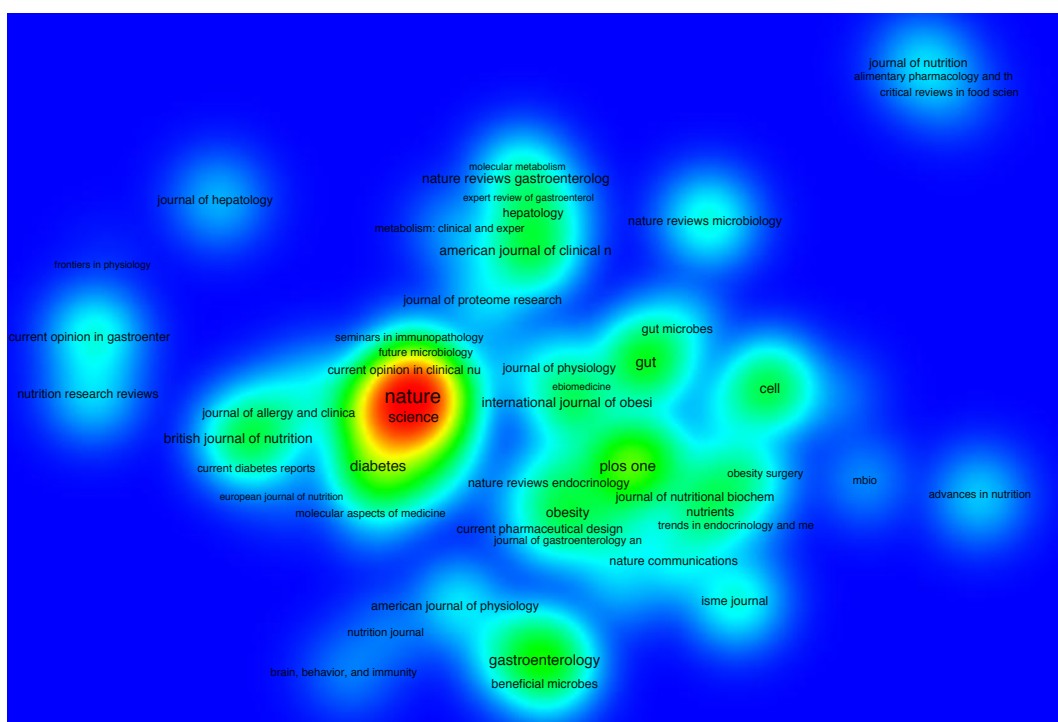

**Figure 6  Density map of journals citation analysis.** One hundred twenty-seven journals were included in this analysis with a minimum productivity of five publications in this field. Journals with the higher number of citations have darker spots.

Modern lifestyles that change the selection pressures on microbiomes could alter exposures to bacteria during the early lives of hosts and thus may contribute to the development of obesity. Antibiotic use in human infancy was significantly associated with obesity development (*Ajslev et al., 2011*). Alterations in the enteric microbiome also occur when interventions are used to treat obesity (*Li et al., 2011*). Thus, modulation of the host's enteric microbiota is a promising way to reduce human obesity (*Li et al., 2017b*; *Zhang et al., 2014*).

Bibliometric study is based on the comprehensive analysis of publications' bibliographic data, such as authors' affiliations, publication types, source countries, funding and citation information (*Pritchard, 1969*). In this study, the significance of intestinal microbiota in the progression and management of obesity has been analyzed using a bibliometric approach. The present study is the first time that a bibliometric overview of academic publications on the topic of intestinal microbiota and obesity has been presented. The percentage of intestinal microbiota related publications in the field of obesity research has been soaring in recent years.

North America and Europe are the most prolific areas in this field; the US and China are the top two most productive countries. Several factors are revealed to be significantly associated with the publication numbers of this field. The most positively related factor is

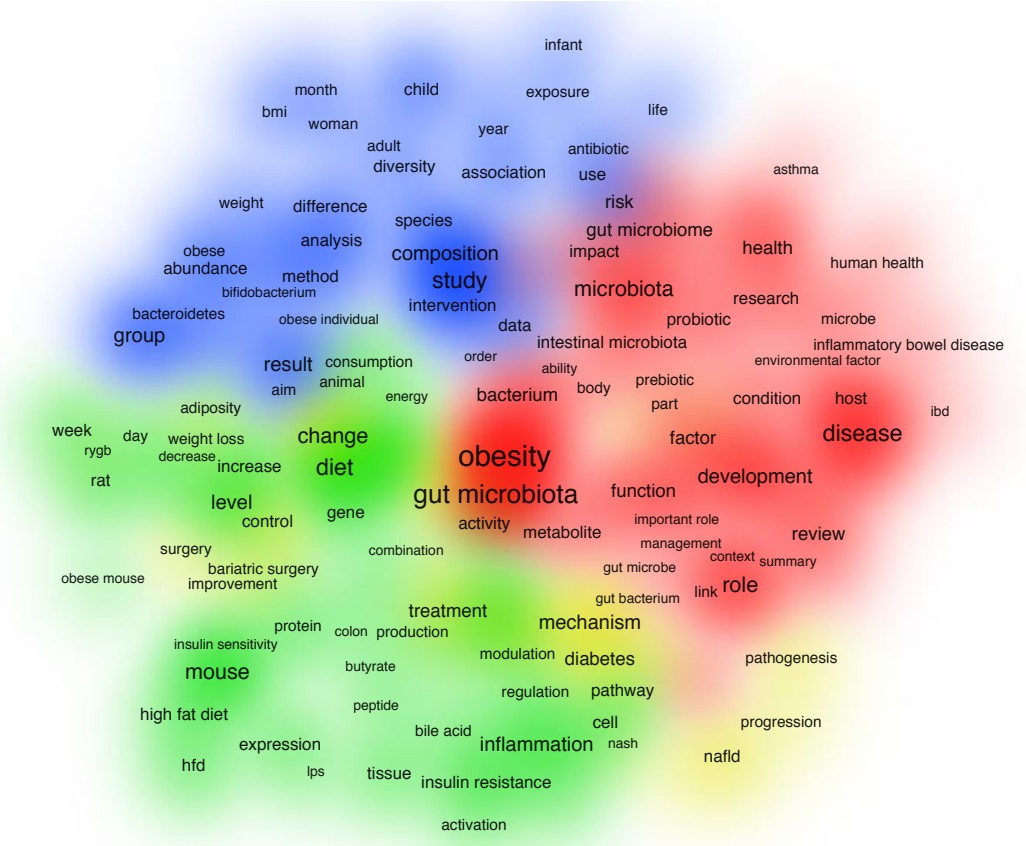

**Figure 7** **Density map of the most frequently encountered terms extracted from the titles and abstracts of retrieved publications.** A number of 195 terms met the threshold with a minimum number of occurrences as 100. The larger circle size or font size indicates higher occurrence.

GDP, which can partly explain the highest productivity of the US. The HDI and PN also show a positive correlation with the publication numbers, but are weaker than GDP.

The most productive journals in this field are *PLOS ONE*, *Scientific Reports,* and *British Journal of Nutrition*. The most productive institutions are *University College Cork* in Ireland, *Inserm* in France, and *Universite Catholique de Louvain* in Belgium. International collaboration analysis also shows that most of the collaborations are between developed countries. However, a multitude of developing countries are also suffering from the increasing prevalence of obesity (*Nasreddine et al., 2017*), so this study reveals the necessity of developed countries to support the developing regions in gut microbiota and obesity-related research in order to address this issue on a global scale.

## CONCLUSIONS

Intestinal microbiota is playing a significant role in obesity research. Data obtained from this study represent the global research trends, collaboration patterns, and spatial density of the role of intestinal microbiota in obesity. These data are helpful for scientific researchers

and public health policymakers in research planning and decision-making of this domain. The current study can also help scientists to locate research hot spots and gaps by offering comprehensive analyses and structured information on this topic.

## ABBREVIATIONS

SCR      standard competition ranking
IF      impact factor

## ACKNOWLEDGEMENTS

The authors would like to thank the University of Chicago for giving us the opportunities to access the most recent information sources such as Scopus database. Our thanks also go to Clara Sava-Segal from the University of Chicago for editing the manuscript.

### Funding

This study was supported in part by grants from National Natural Science Foundation of China Key Program (81730112), China Postdoctoral Science Foundation (2017M610830), National Natural Science Foundation of China (81603378), Natural Science Foundation of Jiangsu Province (BK20160545), and NIH/NCCAM AT004418 and AT005362. There was no additional external funding received for this study. The funders had no role in study design, data collection and analysis, decision to publish, or preparation of the manuscript.

### Grant Disclosures

The following grant information was disclosed by the authors:
National Natural Science Foundation of China Key Program: 81730112.
China Postdoctoral Science Foundation: 2017M610830.
National Natural Science Foundation of China: 81603378.
Natural Science Foundation of Jiangsu Province: BK20160545.
NIH/NCCAM: AT004418, AT005362.

### Competing Interests

The authors declare there are no competing interests.

### Author Contributions

- Haiqiang Yao performed the experiments, analyzed the data, contributed reagents/materials/analysis tools, prepared figures and/or tables, authored or reviewed drafts of the paper.
- Jin-Yi Wan performed the experiments, analyzed the data, prepared figures and/or tables, authored or reviewed drafts of the paper.
- Chong-Zhi Wang, Wei-Hua Huang and Jinxiang Zeng prepared figures and/or tables.
- Lingru Li, Ji Wang and Yingshuai Li analyzed the data.

- Qi Wang conceived and designed the experiments, contributed reagents/materials/analysis tools, approved the final draft.
- Chun-Su Yuan authored or reviewed drafts of the paper.

## Data Availability

The raw data has been provided as a Supplemental File.

## Supplemental Information

Supplemental information for this article can be found online at http://dx.doi.org/10.7717/peerj.5091#supplemental-information.

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
