# Peer review of "Bibliometric analysis of research on the role of intestinal microbiota in obesity"

_PeerJ, doi:10.7717/peerj.5091_

## Round 0.1 · original submission · Major Revisions

Dear Dr. Yao,

I have now received reports from two reviewers who are generally supportive of publication but they have raised concerns that preclude acceptance of the manuscript in its current form. Accordingly, I invite you to respond to the reviewers' and editor's comments and recommendations below. In particular note Reviewer 2's suggestions that you could go deeper (and make the article more significant) by pushing the analysis further.

The revised version will be re-reviewed and a decision on acceptability of your manuscript will be made only after the revised version has been re-evaluated. Please understand that this invitation to resubmit a paper is not accompanied by a commitment to publish, since a revision still may not achieve a sufficient priority to warrant acceptance.

If you decide to resubmit the revised version, please summarize all the improvements made in the new version and give answers to all critical points raised in the reviewers’ report in an accompanying letter. I strongly suggest deepening the data analysis according to the reviewers' suggestions and reconsidering the selection criteria for articles to be included in the review.

Regards,

Stefano Menini

Reviewer 1 ·

Basic reporting

Clear and unambiguous, professional English used throughout
Literature references, sufficient field background/context provided
Professional article structure, figs, tables. Raw data shared.

The paper generates should include following kinds of data (Especially fort the Results section
*Did you investigate correlations were between the number of publications
about “gut microbiata” and population number (PN), gross domestic
product (GDP), gross domestic product per capita (purchasing
power parity, PPP), gross domestic product per hour worked
(GDP2), human development index (HDI), Internet users
(IU), percentage of individuals using the Internet (PIUI; according
to total population of a country), EF English
Proficiency Index (EF EPI), and productivity (P) development
indicators of the countries.This will make your work more unique
*Citations analysis of active organizations in
publishing articles can be done

Experimental design

Original primary research within Aims and Scope of the journal.
Research question well defined, relevant & meaningful. It is stated how research fills an identified knowledge gap.

Rigorous investigation performed to a high technical & ethical standard.

In methods section:
* The keywords regarding obesity were: obesity, bariatrics, corpulence, fatness, and overweight. bariatrics is a definition of obesity treatment not obesity.I think that you should be removed from the keywords.
* why you should use the SCOPUS database more clearly. We know that, WoS database is the most reliable service for publications and citations.

Validity of the findings

*The discussion was only made up of the introduction paragraphs and the repetition of the results.Present results should be evaluated and discussion should be rewritten.
The introduction paragraphs should be shortened.

Additional comments

A very up-to-date and valuable topic was discussed in the study. But the data must be expanded a bit and the discussion should be rewritten in the light of this data.
You can see the article in below for an example:
Ozsoy Z, Demir E. The Evolution of Bariatric Surgery Publications and Global Productivity: A Bibliometric Analysis. Obes Surg. 2017 Oct 31. doi: 10.1007/s11695-017-2982-1. [Epub ahead of print]

Reviewer 2 ·

Basic reporting

INTRO
This study makes a bibliometric analysis on the topic of intestinal microbiota in obesity. Authors find that this field of research is rapidly growing, that English is the most common language, USA the most prolific country, most prolific journals and authors are identified.

BASIC REPORTING
The paper is well written and the context of the study is well described in the introduction paragraph.

Structure of the study is consistent with journal standards.

Figures are high quality. In our opinion, labels should be reconsidered. Indeed, labels should rather introduce the figure than make a comment of it or describe the methods. For instance, in figure 2, the sentence “Gordon J. I. ranked the first with the citations of 21177”, could better appear in the text and not in the label; in figure 3, the text “by using ArcMap 10.1 software” should appear rather in the methods section. And so on.
Palette colors of figure 3 should be reconsidered. The color nuances should represent the degree of the measure, if you radically change color for each category, the message is much less clear. Try to consider moving from color 1 to color 2 with different nuances corresponding to different categories.

Raw data are supplied.

Experimental design

EXPERIMENTAL DESIGN
The topic is original and relevant for the journal.

Methods are clear with sufficient detail to replicate.

We think that authors should better explain bibliometric indicators used in the paper. In fact, this is not a bibliometric journal, and the auditory may not be familiar with this field. We think that it would be of great help to better explain Price’s and Lotka’s law, etc.

We think that this bibliometric analysis would benefit from a better contextualization. This means that data should be reframed in a larger context, for instance: the growth of scientific publications should be compared to the general scientific production (or the scientific production on obesity, or whatever could be relevant as a frame).
Same thing for the number of papers per country. In whatever fields, USA would be probably first for the absolute number of publications, followed by a bunch of the same countries close to each other. So this is of little interest. We think that data should be weighted to better reflect the real interest of a national scientific community on a topic. Data could be weighted on the national population or, in this case, on obesity prevalence (or else).

If data 2017 are not complete, they should be omitted.

Please note that rounding at the second decimal does not add any capital information and make the text heavier to read.

Application of the H-index to articles and not to a researcher is unusual. This indicator was not conceived to describe top-rated papers. Please reconsider the paragraph.

The distribution of citation per paper is not gaussian (but rather hyperbolic), so reporting the mean citations per paper has little interest and, on the contrary, can be misleading, as that could suggest that each paper has 37 citations, which is false. Please reconsider the paragraph.

We think that in bibliometric analysis a compromise should be found between comprehensiveness and “noise”. Authors include papers from any language, any type of publication (even conference papers, notes, letters etc…), any journal in any field. They want to be comprehensive. The counterpart is that they retrieve a great number of papers of a questionable interest. For instance, they included papers in other language than English (which will probably have only a local public), conference papers that are available on Scopus (which is not freely accessible) but not on Pubmed (which is freely accessible), and so on. We think that these criteria add a “background noise” to the study.

Validity of the findings

VALIDITY OF THE FINDINGS
This is the first bibliometric study on intestinal microbiota in obesity, by consequence results are original.
Authors apply clear and well-known bibliometric indicators to data.
We think that criteria of inclusion (as discussed above) are too large, so that could possibly
make the results less reliable.
Conclusion are well stated.

Additional comments

GENERAL COMMENTS
This study is well written. Graphics is high quality. Topic is interesting. No major flaw is present. Bibliometric indicators could be better explained. Selection of articles is questionable (“background noise”). Analysis could go deeper.

---

## Round 0.2 · Minor Revisions

Dear Dr. Yao,

Thank you for your resubmission. I have now received reports from our reviewers who are generally supportive of publication. However, reviewer 2 suggested minor modifications. Accordingly, I invite you to address the reviewer' s 2 comments and recommendations, and follow their valuable suggestions

With kind regards,

Stefano Menini

Reviewer 1 ·

Basic reporting

no comment

Experimental design

no comment

Validity of the findings

no comment

Additional comments

no comment

Reviewer 2 ·

Basic reporting

This is a second reviewing after corrections.
Authors considered most but not all reviewers’ remarks. The overall quality of the manuscript has increased.
The paper is well written and the context of the study is well described in the introduction paragraph.
Structure of the study is consistent with journal standards.
Authors have modified figures as suggested.
Raw data are supplied.

Experimental design

Authors calculate the growth trend of accumulated documents. The trend of publication growth has been theorized by Price theory and is a weel-established bibliometric indicator. Nevertheless the exponential growth should not be calculated on the accumulated number of documents, but on the absolute number of documents per year. The reported equation on accumulated documents has no interest and is not a commun bibliometric indicator. Please check this for instance in García-García P, Eur J Obstet Gynecol Reprod Biol. 2005

Concerning the number of publications per country, authors considered reviewers remarks. Nevertheless they do not report (event in supplemental data) the adjusted country ranking (adjusted for GDP or/and inhabitants number or else). Please check Daby Y, Obes Surg 2016

Validity of the findings

This is the first bibliometric study on intestinal microbiota in obesity, by consequence results are original.
Authors apply clear and well-known bibliometric indicators to data.
Conclusions are well stated.

Additional comments

Authors improved the manuscript quality after initial reviewing.

---

## Round 0.3 · accepted · Accept

Dear Dr. Yao,

Our referees have now considered your paper and have recommended publication in “PeerJ”. We are pleased to accept your paper in its current form which will now be forwarded to the publisher for copy editing and typesetting.

I thank all reviewers for their effort in improving the manuscript and the authors for their cooperation throughout the review process

Yours sincerely,

Stefano Menini

# Reviewer 2 ·

Basic reporting

Third round of reviewing. Manuscript has improved

Experimental design

Corrections were done according to reviewers' remarks.

Validity of the findings

Consistent

Additional comments

Final version is suitable for publication .